# Enzyme-Based Biostimulants Influence Physiological and Biochemical Responses of *Lactuca sativa* L.

**DOI:** 10.3390/biom13121765

**Published:** 2023-12-09

**Authors:** Rachele Tamburino, Teresa Docimo, Lorenza Sannino, Liberata Gualtieri, Francesca Palomba, Alessio Valletta, Michelina Ruocco, Nunzia Scotti

**Affiliations:** 1Istituto di Bioscienze e BioRisorse (CNR-IBBR), 80055 Portici, Italy; rachele.tamburino@ibbr.cnr.it (R.T.); teresa.docimo@ibbr.cnr.it (T.D.); lorenza.sannino@ibbr.cnr.it (L.S.); 2Istituto per la Protezione Sostenibile delle Piante (CNR-IPSP), 80055 Portici, Italy; liberata.gualtieri@ipsp.cnr.it (L.G.); francesca.palomba@ipsp.cnr.it (F.P.); michelina.ruocco@ipsp.cnr.it (M.R.); 3Department of Environmental Biology, Sapienza University of Rome, 00185 Rome, Italy; alessio.valletta@uniroma1.it

**Keywords:** biomolecules, growth elicitors, lettuce, antioxidant activity, sustainable agriculture, nutraceuticals

## Abstract

Biostimulants (BSs) are natural materials (i.e., organic or inorganic compounds, and/or microorganisms) having beneficial effects on plant growth and productivity, and able to improve resilience/tolerance to biotic and abiotic stresses. Therefore, they represent an innovative alternative to the phyto- and agrochemicals, being environmentally friendly and a valuable tool to cope with extreme climate conditions. The objective of this study was to investigate the effects of several biomolecules (i.e., Xylanase, β-Glucosidase, Chitinase, and Tramesan), alone or in combinations, on lettuce plant growth and quality. With this aim, the influence of these biomolecules on biomass, pigment content, and antioxidant properties in treated plants were investigated. Our results showed that Xylanase and, to a lesser extent, β-Glucosidase, have potentially biostimulant activity for lettuce cultivation, positively influencing carotenoids, total polyphenols, and ascorbic acid contents; similar effects were found with respect to antioxidative properties. Furthermore, the effect of the more promising molecules (Xylanase and β-Glucosidase) was also evaluated in kiwifruit cultured cells to test their putative role as sustainable input for plant cell biofactories. The absence of phytotoxic effects of both molecules at low doses (0.1 and 0.01 µM), and the significantly enhanced cell biomass growth, indicates a positive impact on kiwifruit cells.

## 1. Introduction

Modern agriculture practices need to reduce the use of chemical inputs while satisfying the increasing demand for food due to a growing world population. Concomitantly, climate changes expose crops to more frequent and extreme environmental conditions (e.g., drought, heat, and flooding) and strongly affect crop yield. Indeed, agriculture is the sector mostly under pressure and, so far, new solutions and approaches need to be embraced globally to ensure food production at the desired level of quantity and quality, and to enhance resource use efficiency. The selection of more tolerant varieties through breeding, genetic engineering, and marker-assisted selection (MAS) is expensive and time-consuming [1]. Therefore, the search for innovative solutions, and environmentally friendly and sustainable farming practices represents an integral part of modern agriculture. In this context, biostimulant (BS) substances defined by Regulation (EU) 2019/1009 as “product stimulating plant nutrition processes independently of the product’s nutrient content with the sole aim of improving nutrient use efficiency, tolerance to (a) biotic stress, quality traits, availability of confined nutrients in soil or rhizosphere” represent an innovative and valuable tool. BSs are classified according to their nature into seaweed and plant extracts, protein hydrolysates, humic substances, inorganic compounds, and microorganisms. To date, seaweed extracts (SEs) are the most studied and commercialized BSs [2,3]; however, in a circular economy perspective, protein hydrolysates (PHs) are gaining increasing attention as they usually derive from hydrolysis of wastes of agricultural or animal processing [1]. In line with sustainable farming practices, plant proteins or plant-based proteins are preferable over animal by-products because in terms of energy use and environmental impact they have a positive balance. PHs are mixtures of free amino acids, polypeptides, and oligopeptides whose proportions may change depending on the hydrolysis method used [1]. Small peptides and amino acids exhibit phytohormone-like activities, potentially stimulating root growth to enhance nutrient absorption. They also modulate carbon and nitrogen metabolism by increasing enzyme activity in the tricarboxylic acid cycle (TCA), and enhance antioxidative properties by accumulating antioxidant compounds such as phenolics, ascorbic acid, and carotenoids [4]. The latter characteristic, helping plants to cope with stress, also reflects in the boost of nutraceutical quality of the products. The application of PHs has beneficial effects that have been reported for a great variety of plant species including maize, tomato, olive, cabbage, and lettuce [5,6,7,8,9,10,11,12,13].

The development of new highly active plant BSs has become an important focus of research interest in the context of climate changes and the ecological transition, already underway in several countries. Indeed, the use of BSs requires the application of minimal quantities [14,15], and this helps maintaining the agro-ecological equilibrium which is instead jeopardized by the massive use of pesticides and chemical fertilizers in the usual agricultural practices. Aside from protein hydrolysate, to the best of our knowledge, to date, enzymes have never been tested for their potentiality as elicitors of plant growth and resilience. Therefore, in the attempt to identify new and effective BSs, here, we investigated the suitability of two plastid-based proteins usually used for lignocellulosic bioconversion into fermentable sugars, namely a thermophilic Xylanase from *Alicyclobacillus acidocaldarius* and a hyperthermophilic β-Glucosidase from *Pyrococcus furiosus* [16], and a Chitinase, known as a biocontrol enzyme being able to inhibit or kill fungi and insects [17,18,19], as biostimulants on lettuce plants (*Lactuca sativa* L.). As is known, these enzymes are being used in various industrial application ranging from food and feed processing to pharmaceuticals [20]. Although some of them, including Cellulases and Xylanases, are commercially available in purified form or through microbial production, their production costs are very expensive. Therefore, in our study, we used plant plastids as an economic, safe, and easily scalable system for production of β-Glucosidase and Xylanase enzymes, and tested their activity as novel biostimulants. In detail, we compared the effects of thirteen treatments, including our plastid-based proteins and a commercial preparation of Chitinase alone or combined with the polysaccharide Tramesan [21]. *Lactuca sativa* L. was chosen because it is one of the most widely consumed leafy vegetable worldwide and represents a significant source of antioxidants and bioactive compounds [22,23]. With this aim, we evaluated the effects of the enzyme-based biostimulants on plant biomass growth and the contents of health-promoting compounds such as carotenoids and phenolic substances. Moreover, to envisage novel sustainable solutions to improve growth and related productivity of cells biofactories, we tested the more promising molecules on kiwifruit cell lines to evaluate any positive impact on cell viability and growth.

## 2. Materials and Methods

### 2.1. Plant Material and Growth Conditions

The experiments were carried out in two subsequent years in a greenhouse during spring–summer seasons (May–June or June–July). Three week old crisphead type lettuce (*Lactuca sativa* L.) seedlings, iceberg variety, were provided by a local plant nursery in a nursery tray. After transplanting, plants were placed in a greenhouse under natural light and temperature conditions (15 h daylight/9 h dark period, T_min_ = 25 °C, T_max_ = 35 °C). Lettuce plants were irrigated daily using tap water (200 mL); on the day of biostimulant application, plants were irrigated with tap water (150 mL); BSs (50 mL) were applied at least 1 h after irrigation; then, plants were further irrigated after 24 h.

### 2.2. Preparation of Biostimulant Solutions and Lettuce Plant Treatments

BS solutions were prepared by dilution of the selected biomolecules at a concentration of 1 µM in PBS 1X based on previous optimization experiments (Table 1).

Xylanase from *A. acidocaldarius* and β-Glucosidase from *P. furiosus* were extracted from transplastomic tobacco plants as described by Castiglia et al. [16].

Chitinase (Chimax-N) was purchased from Amicogen Inc. ( Jinju-si, Republic of Korea).

Tramesan [24] was kindly provided by Professor Massimo Reverberi, Department of Environmental Biology, Sapienza University of Rome, Italy.

Six days after transplanting (DAT) plants were irrigated with control solution (S1) and biostimulant ones (S2–S13). Treatments were applied every seven days for three weeks to a total of 195 lettuce plants (15 plant replicates for each treatment), arranged in a randomized design. Seven days after the last treatment (27 DAT), lettuce plants were harvested for biomass determination and biochemical analyses (Figure 1A).

### 2.3. Chlorophyll Content

During the time frame of the experiment (three weeks), chlorophyll content on lettuce plants from seedling stage to the end of the experiment was non-destructively determined using the portable Soil Plant Analysis Development chlorophyll meter (SPAD 502, Konica Minolta, Osaka, Japan). The detected SPAD index is proportional to the concentration of chlorophyll present in the leaf, and is based on the absorbance in the red and near-infrared regions [25]. Readings were taken weekly, three days after each BS application around 10 AM to avoid any reduction in photosynthetic capacity, on the two fully expanded leaves per plant for each biological replicate (*n* = 15). For each treatment, SPAD values are expressed as mean ± SD of 15 biological replicates. 

### 2.4. Plant Growth and Biomass Determination

After 7 days from the last treatment application, plants were harvested and both fresh and dry weights were measured. Fresh weight values are expressed as mean ± SD of 10 biological replicates. Dry weight measurement was performed for each plant by placing their aerial part in a drying oven at 105 °C until constant weight was reached, then the dry weight of leaves was separately measured. For each treatment, dry weight values are expressed as mean ± SD of 10 biological replicates.

### 2.5. Determination of Pigment Content of Lettuce

To assess the effects of BSs application on the pigment content of the plants, total chlorophyll and carotenoids were extracted by homogenization of fresh leaf tissues (0.1 g) in 96% ethanol (*v*/*v*) containing 0.3 mg/mL NaHCO_3_. The resulting extracts were centrifuged at 13,000× *g* for 5 min at 4 °C. Samples were kept on ice in the dark during the experiment. Chlorophyll a (Chl a), chlorophyll b (Chl b), and carotenoid (Car) contents were determined by taking absorbance of the supernatant at 665, 649, and 470 nm, respectively by a UV–Vis spectrophotometer MultiskanTM Sky microplate spectrophotometer (Thermo Scientific, Waltham, MA, USA) following the method described by [26]. Results are expressed as µg/100 mg FW (means ± SD of 5 biological replicates).

### 2.6. Protein Extraction and Quantitation

Total soluble proteins were extracted by homogenization in 0.1 M Tris-HCl pH 7.8 containing 0.2 M NaCl, 1 mM EDTA, 0.2% Triton X100, 2% SDS, 2% β-mercaptoethanol, 1 mM PMSF, and 1X proteinase inhibitor cocktail. Samples were centrifuged at 20,000× *g* for 15 min at 4° C. Supernatant was recovered and proteins were quantified by Bradford method using Biorad protein assay reagent (Biorad, Hercules, CA, USA). Bovine serum albumin (BSA) was used as reference standard (range 2–10 µg/mL, 4 levels, R^2^ = 0.999). Results were expressed as mg/g FW (means ± SD of 5 biological replicates).

### 2.7. Determination of Antioxidant Compounds of Lettuce and Antioxidant Activity

Total phenolic content (TPC) was determined by the Folin–Ciocalteu method according to [27]. Frozen powdered leaf tissues (250 mg) were extracted in 1 mL of 80% methanol (*v*/*v*) incubated in agitation overnight in the dark; 20 μL of extract was mixed with 150 μL of the Folin–Ciocalteu reagent diluted with distilled water (1:30, *v*/*v*) and 30 μL of 20% (*w*/*v*) Na_2_CO_3_. After 45 min in the dark at room temperature, the absorbance at 765 nm was determined in a MultiskanTM Sky microplate spectrophotometer (Thermo Scientific, Waltham, MA, USA). Gallic acid (GA) was used as reference standard and TPC was estimated from the GA calibration curve (range 5–200 µg/mL, 7 levels; R^2^ = 0.999). The results were expressed as GA equivalents per 100 g of fresh leaf (mg GAE/100 mg FW, means ± SD of 5 biological replicates).

Ascorbic acid content was determined as reported by [28]. Briefly, 0.5 g of lettuce frozen powder were added to 300 μL of ice-cold 6% trichloroacetic acid (TCA). Samples were vortexed and kept on ice for 15 min, then centrifuged at 25,000× *g* for 15 min at 4 °C. The reaction samples were then prepared by mixing 20 μL of supernatant with 20 μL of 0.4 M phosphate buffer (pH 7.4), 10 μL of double distilled H_2_O, and 80 μL of reagent solution, composed of solution A (31% H_3_PO_4_, 4.6% (*w*/*v*) TCA and 0.6% (*w*/*v*) FeCl_3_) and solution B (4% 2,20-dipyridil (*w*/*v*) in 70% ethanol) at a proportion of 2.75:1 (*v*/*v*). The mixture was incubated at 37 °C for 40 min and measured at 525 nm by using a MultiskanTM Sky microplate spectrophotometer (Thermo Scientific). Sodium ascorbate (NaAs) was used as reference standard by preparing series dilutions of 0, 5, 10, 20, 30, 40, and 50 nmol/20 µL in 6% TCA. Reduced ascorbate (AsA) was estimated from the calibration curve (range 0.025–0.15 nmol/µL, 5 levels; R^2^ = 0.997). Five biological replicates for each sample and three technical assays for each biological repetition were measured. Values were expressed as mg/100 g FW (means ± SD of 5 biological replicates).

The 2,2′-azino-bis (3-ethylbenzothiazoline-6-sulfonic acid) (ABTS) scavenging capacity assay was carried out in 96-well plates according to [27]. Briefly, 500 μL of ABTS^•+^ solution (1 mM) was mixed with 5 μL of diluted extracts (five dilutions with approximately 20–80% control absorbance), Trolox (range 2–25 μM, 6 levels) reference standard solutions or PBS (control), and 300 μL of the mixtures was transferred into a 96-well plate. The absorbance was measured at 734 nm, after 60 min of incubation in the dark at 30 °C, with a microplate spectrophotometer reader MultiskanTM Sky (Thermo Scientific). ABTS assay results were expressed as Trolox equivalent antioxidant capacity (TEAC) per 100 mg FW of extract (μmoL TE/100 g FW, mean ± SD of 5 biological replicates).

### 2.8. Treatments of Plant Cell Cultures with Plastid-Based Xylanase and β-Glucosidase

Kiwifruit plant cell cultures were obtained as reported by [29] with slight modifications. Briefly, leaves from yellow-flesh kiwifruit plants (*Actinidia chinensis* cv. Soreli) grown in the field were used for callus induction. Fully expanded leaves, after being immersed in 70% ethanol for 30 sec, were surface sterilized in 1% NaOCl containing a few drops of Tween 20 (Sigma-Aldrich, Milan, Italy) for 15 min and rinsed 3 times in sterile deionized water. Leaf explants of approximately 5 × 5 mm were placed into a culture medium consisted of Murashige and Skoog salts and vitamins [30] supplemented with 5 µM α-naphthaleneacetic acid (NAA), 1 µM zeatin, 3% (*w*/*v*) sucrose, and 0.8% (*w*/*v*) plant agar (pH 5.8). All components of the culture medium were purchased from Duchefa-Biochemie (Haarlem, The Netherlands). The cultures were maintained under a 16:8 h photoperiod at 70 μmol·m^−2^·s^−1^ supplied by cool white fluorescent tubes. The callus subcultured on agarized medium supplemented with 10 µM 2,4-dichlorophenoxyacetic acid (2,4-D) and 1 µM 6-benzylaminopurine (BA) was suspended in a liquid medium of the same composition (biomass/medium 2 g:20 mL). The cultures were maintained in continuous darkness, shaken (100 rpm), and subcultured every 20 days. Treatments with β-Glucosidase and Xylanase (1, 0.1 or 0.01 µM) were carried out on day 10 of culture, and the impact on viability and cell biomass growth was evaluated on day 20 of culture. Cell viability was assessed using fluorescein diacetate assay (FDA viability-test), as described by [31]. Cell biomass growth was measured after separating the cells from the medium by vacuum filtration and drying the biomass to a constant weight in an oven at 70 °C. The growth of the cells was expressed as relative growth calculated from (Wt − Wi)/Wi, where Wi and Wt are initial and total dry weight, respectively.

### 2.9. Statistical Analysis

The analysis of variance (ANOVA) of the collected data was performed using the SigmaPlot 12 software (Systat Software, Inc., San Jose, CA, USA) and the multiple pairwise comparisons assessed by the Tukey test (*p* value < 0.05, *n* = 5 and *n* = 10).

## 3. Results

### 3.1. Plant Growth and Biomass

Lettuce plants were grown for 27 days corresponding to the complete head maturation (Figure 1A). Control (S1) and BS solutions (S2–S13) whose composition is reported in Table 1 were applied weekly.

Throughout the treatment duration, as illustrated in Figure 1B, no symptoms of toxicity were observed, such as leaf drop or yellowing. In fact, upon visual inspection, it was evident that the treated lettuce plants, when compared to the S1 control group, displayed various positive effects on their appearance and supported healthy growth. Notably, S2 (Xylanase) and S8 (β-Glucosidase) treatments contributed to improved plant appearance, characterized by increased head volume, crispness, and leaf thickness.

To assess whether BS applications can affect the lettuce physiological state, chlorophyll content, closely related to the nutritional and healthy conditions of the plants, SPAD non-destructive measurements were determined (Figure 2). The recorded mean SPAD index indicated that almost all the treatments positively affected the chlorophyll amount, except for S2, S6, and S12 treatments which did not cause significant variations in the chlorophyll content compared to control plants (S1).

The effect on the growth of lettuce plants was monitored at the end of the treatment by measuring fresh and dry weights of BS-treated plants compared with control (S1 treatment). After the last application of BS solution, 10 biological replicates for each condition were used to determine the fresh and dry weights. Fresh weight greatly varied among lettuce plants subjected to the different treatments. In particular, the treatments containing Xylanase (S2), β-Glucosidase (S8), and Xylanase combined with β-Glucosidase (S3) induced a statistically significant increase in biomass up to 35%, whereas all other treatments resulted in a decrease in biomass up to 17% (Figure 3A). Consistent with fresh weight measurements, S2, S3, and S8 plants showed the highest increases in their dry weight as compared with controls (S1) with improvements of 28%, 37%, and 21%, respectively, whereas all the other treatments did not significantly vary as compared to controls (Figure 3B). Overall, these results indicate a positive influence of both Xylanase and β-Glucosidase in enhancing plant biomass.

### 3.2. Pigment Content of Lettuce after Treatments

Considering the limitations of the non-destructive method for the determination of absolute chlorophyll content, in addition to SPAD index, the concentration of photosynthetic pigments was measured as a valuable indicator of the plant physiological state. Among the treatments, only the Xylanase-based one (S2 treatment) induced a significant increase in carotenoids (up to 63%) and a simultaneous decrease in chlorophyll b content (up to 56%) with consequent alteration of chlorophylls and chlorophylls/carotenoid (Chls:Car) ratios. Indeed, in S2 treated plants, the Chl a:b ratio was three times higher than in other treatments (i.e., 6 in S2 vs 2–2.8 in other treatments); conversely, the Chls:Car ratio was 2–3 times lower than in other treated plants (i.e., 4.5 in S2 vs. 8.2–11.2 in other treatments) (Figure 4).

### 3.3. Antioxidant Compounds of Treated Lettuce

To assess whether the enzyme-based biostimulants might boost the accumulation of polyphenols, the total phenolic content (TPC) was evaluated using the Folin–Ciocalteu colorimetric assay. A peculiar positive effect of Xylanase alone (S2) (0.82 mg GAE/100 mg FW) was observed, resulting in a significant increase of 62% in total phenol content as compared to the S1 treatment (Figure 5), although, to a lesser extent, the S3 (Xylanase combined with β-Glucosidase) and S8 (β-Glucosidase) treatments also positively affected polyphenol content as compared to the other treated and control plants. Interestingly, when Xylanase and β-Glucosidase were combined either with Chitinase or Tramesan (S4-S5-S6-S10-S12), treatments had a negative effect on polyphenol accumulation, being even lower than control plants. In particular, the combination of Xylanase, β-Glucosidase, and Chitinase molecules (S7) resulted in the lowest polyphenol levels reaching a 54% significant decrease in TPC of treated lettuce as compared to control. All other treatments did not show significant differences in TPC (S9-S11-S13) as compared with control.

Based on the TPC results and considering that ascorbic acid (AsA) contributes about 24.5% of the total antioxidant activity in lettuce [32,33], we investigated whether some of the treatments that mostly impacted on TPC, either positively or negatively, would have similar effects in terms of AsA content. With this aim, four different treatments were selected, namely, S2, S3, S7, and S8, and monitored for their AsA-induced accumulation as compared to control. Significant changes were observed in the reduced form of ascorbic acid (AsA) concerning the application of the BSs (Figure 6). In particular, Xylanase and β-Glucosidase applied separately (S2 and S8, respectively) or in equimolar combination (S3) had a clear positive effect on AsA accumulation that increased up to 28% as compared to the control. Conversely, when both molecules were combined with Chitinase (S7 solution) the AsA accumulation reduced by 36% as compared to plants under control conditions confirming negative effects for the S7 treatment.

### 3.4. Antioxidant Activity of Treated Lettuce

To verify whether there was a correlation between the boosting effects of selected treatments on both TPC and AsA content, and the antioxidant activity of treated plants, an ABTS assay, based on the reduction of the ABTS^•+^ radical action by the antioxidants in the sample, was performed on S1, S2, S3, S7, and S8 selected treatments. The levels of antioxidant capacity reflected the trend of accumulation observed for total polyphenols (Figure 7). Indeed, S2 and S8 plants treated with Xylanase and β-Glucosidase, respectively, showed a significantly higher antioxidant power evaluated by the ABTS^•+^ scavenging activity with a TEAC value of almost 140 µmmol/g FW as compared to plants either under control conditions or exposed to S3 or S7 treatments whose content was about 100 and 94 µmmol/g FW, respectively, and not significantly different among samples.

### 3.5. Protein Content

Total soluble proteins in S1, S2, S3, S7, and S8 treated lettuce plants following the application of enzyme-based BSs was also determined. A significant increase was observed only in S3-treated lettuce as compared with S1 control plants (Figure 8), suggesting that proteins likely coincide with the dry weight increase detected in these treated plants.

### 3.6. Treatment of Plant Cell Cultures with Xylanase and β-Glucosidase

Considering the importance of plant cell cultures in basic and applied research [34] and the effects of Xylanase on lettuce plants, kiwifruit cell lines were used to investigate their effects on cell growth and viability. To choose the right concentration for the viability assay, both enzymes were tested at three concentrations and their effects were compared to their respective controls. Figure 9A,B show that both Xylanase and β-Glucosidase at the highest dose (1 µM) showed a toxic effect on cells by reducing plant cell viability by almost 50 and 40%, whereas at lower doses (0.1 and 0.01 µM) no negative effect on cell viability was detected.

The assessment of the effects of both Xylanase and β-Glucosidase on plant cell biomass (Figure 9C,D) showed a negative effect at the highest dose of both enzymes, similarly to that observed for cell viability, whilst a positive effect on cell proliferation was observed for both enzymes at the lower tested concentrations. In particular, Xylanase-treated cells showed an increase of 50 and 40% (i.e., 1.5 and 1.4) at 0.1 and 0.01 µM, respectively, whereas β-Glucosidase induced a biomass accumulation of 60 and 30% (i.e., 1.6 and 1.3) at 0.1 and 0.01 µM, respectively. Thus, they were revealed to enhance plant cell biomass accumulation but not in a dose dependent manner.

## 4. Discussion

Biostimulants are substances or materials other than nutrients and pesticides that can be used to regulate the physiological processes in plants to stimulate their growth. Therefore, BSs are considered environmentally sustainable and economically favourable answers to optimizing crop productivity [35]. Over the last decade, the BS market greatly increased; thus, the identification and/or the development of new BS products that may enhance plant vegetative growth and its nutritional quality is gaining attention [36]. In particular, their wide open definition allows the investigation of any kind of molecule for its BS properties. With this aim, here, we evaluated, for the first time, the suitability of two plastid-based proteins (i.e., thermophilic Xylanase from *A. acidocaldarius* and hyperthermophilic β-Glucosidase from *P. furiosus* [16]) and a commercial Chitinase, alone or in combinations with Tramesan, as plant growth promoters of lettuce plant growth and elicitors of bioactive molecules such as polyphenols. Xylanase, β-Glucosidase, and Chitinase are microbial enzymes able to hydrolyse complex polysaccharides, usually used for hydrolysis of lignocellulosic biomass [37,38]; therefore, it is worth also envisaging a BS effect on plants by supporting the growth of beneficial rhizobacteria [20].

None of the treatments applied showed any toxicity such as leaf dropping or yellowing on lettuce plants. Overall, based on the measurements performed, the Xylanase and β-Glucosidase containing solutions (S2 and S8) resulted in the most promising treatments. They both conferred a better appearance to treated plants that were improved in their aspect looking healthier than lettuce under other treatments. The non-destructive evaluation of chlorophyll content revealed SPAD values similar to or higher than controls for almost all treated plants confirming the absence of toxicity and a good performance of the photosynthetic machinery for all the solutions utilized. Among all the treatments, fresh and dry weights of S2-, S3-, and S8-treated plants indicated a significant positive influence on plant biomass growth (up to 35 and 37%, respectively). As recently demonstrated by [39], BS effects on lettuce plant biomass yield is cultivar-specific. Indeed, following foliar application of vegetal-derived protein hydrolysates, “Ballerina” cv recorded 51% higher yield than control, whilst “Canasta” showed a modest 12.5% increase. Quantitative evaluation of pigments revealed that Xylanase was the only BS solution that significantly impacted on chlorophyll and carotenoid content resulting in altered Chl a:b and Chls:Car ratios. The decrease in Chl b may be due to its conversion into Chl a, via the so-called “chlorophyll cycle” [40] to optimize adaptation of plants to varying light conditions [41,42,43,44]. The chlorophyll a:b ratio is a highly stable value that is generally used as a parameter to monitor plant oxidative stress status/tolerance because it is decreased by oxidative damage, according to its severity [45]. Both chlorophylls are degraded by oxidative stress; however, it was found that chlorophyll a is more easily degraded by oxidative stress than chlorophyll b [46]. Hence, the elevated Chl a:b ratio detected in Xylanase-treated plants (S2) indicates enhanced stress tolerance. Nevertheless, given the stable levels of Chl a alongside an increase in Car, it is more plausible that Chl b is either degraded or transformed into carotenoids. This assumption is based on the shared precursor of both chlorophylls and carotenoids, namely GGPP (geranyl geranyl pyrophosphate) [43]. The rise in carotenoid content has been commonly reported in plants treated with several types of BSs [47,48]. Indeed, a similar accumulation of carotenoid content (+74%) was previously described for cucumber fruit after foliar application of humic-acid-based solution [49], whereas baby leafy lettuce displayed 16% and 11% increases in carotenoid content when treated with seaweed extracts and protein hydrolysate, respectively [50]. Carotenoids are key pigments for plant response to oxidative stress as they are involved in photoprotection [51]. Therefore, their enhancement supports the hypothesis of an improved stress tolerance in S2-treated plants.

Green leafy vegetables are known to be rich in antioxidants and their levels might be properly increased for added benefits. In this study, Xylanase and β-Glucosidase, alone or in combination (S2, S3 and S8 treatments), were shown to be elicitors of antioxidant health-promoting molecules such as phenolic compounds and ascorbic acid, that were increased by up to 62% and 28%, respectively. Similarly, seaweed-treated spinach showed a concomitant increase in phenolics and AsA of 31 and 79%, respectively [52], and a 5.6-fold increase in AsA content was observed in lettuce treated with a commercial preparation of protein hydrolysate [41].

Based on our results, Xylanase and β-Glucosidase alone or combined were further investigated for their ability to improve the nutritional value and physiological state of lettuce. Concordantly, the application of Xylanase- and β-Glucosidase-containing solutions by increasing the content of major antioxidant molecules influenced the antioxidant activity of lettuce plants as revealed by the ABTS reduction assay, suggesting that these preparations may further boost the nutritional quality of treated plants. Nevertheless, the combination of these two enzymes as for the S3 treatment does not have an additive or synergistic effect, which might be due either to a dose effect of the enzyme or to a possible overlapping in the pathways activated by the enzymes. This result appears even more evident when three enzymes are combined, as for the S7 treatment, where a clear negative effect, probably due to the presence of Chitinase, is exerted both on TPC and AsA. Previously, the enrichment of carotenoids, ascorbic acid, and other antioxidative molecules has been reported in lettuce treated with seaweed extracts and plant protein hydrolysate under sub-optimal N concentration in the soil [53,54,55]. To further verify the improvement in terms of nutritional quality induced by the applied treatments, total proteins were determined. The association of Xylanase and β-Glucosidase (S3) significantly enhanced protein accumulation as compared to S1 plants possibly coinciding with the increment in dry weight of these plants. Conversely, plants treated with the Xylanase-based solution (S2) as well as the β-Glucosidase preparation (S8) showed a significant decrease in protein content that was not correlated with the trend observed for their dry weight, thus suggesting that these treatments might increase other biochemical characteristics. Increased levels of total soluble proteins were detected in both lettuce leaf tissue (up to 15.3%) and spinach (up to 32%) treated with protein hydrolysate solution and borage extracts, respectively [56,57]. Accordingly, lettuce seedlings supplemented with seaweed extracts displayed 38% increase in total protein content [6]. Several reports showed that seaweed extracts improved the crude protein content in plant families like Fabaceae and Poaceae with the highest increase in protein content by two-fold (~200%) recorded in *Vigna sinensis* [58,59]. As suggested, BS-treated plants often have an increased protein content which may be due to a better capability of absorbing elements, in particular to an efficient uptake of nitrogen which is correlated to protein synthesis [60]. Nevertheless, further studies are necessary to better investigate the mechanism of action of our BSs in relation to nitrogen metabolism.

Despite its role in plant growth, development, and immunity, and its great potential in biocontrol [17,18,19], Chitinase did not show biostimulant effects on the parameters we measured either alone or in combination with other enzymes/Tramesan, since a decrease in TPC, protein, and AsA content was observed.

The effects of Xylanase and β-Glucosidase were also evaluated on the viability and biomass of in vitro kiwifruit cell cultures. At high concentrations (1 µM), both enzymes had inhibiting effects on viability and cell culture biomass, whereas at lower doses they significantly stimulated plant cell biomass growth up to 50 and 40% for Xylanase and 60 and 30% for β-Glucosidase at 0.01 µM and 0.1 µM, respectively. Our data clearly highlight the potentiality of Xylanase and β-Glucosidase in improving plant biomass at both whole plant and cellular level. Indeed, to our knowledge, this study is the first attempt reporting the potential of BSs for plant cell culture growth promotion. Further studies on other plant cell cultures can open the possibility to develop sustainable plant cell biofactories for massive production of value-added molecules, which is an important possibility especially to expand healthcare worldwide, in agreement with the Sustainable Development Goals (SDGs) of the United Nations (https://www.un.org/sustainabledevelopment/ (accessed on 3 November 2023)).

It is worth noting that outcomes from BS applications vary depending on a range of factors such as the plant species or varieties, the dosage used, and the mode and the time of application [4]; therefore, it is conceivable that, having a different absorption ability, in vitro cell cultures may be exposed differently in terms of mode, dosage, and time with respect to the whole plant even using the same BS dose in the two systems as we did.

Regarding their mechanism of action, we may assume that both Xylanase and β-Glucosidase have a positive impact on plant growth as the following:Whole proteins; through their enzymatic activities they release sugars that are made available to soil microorganism as an energy source. Indeed, this is the reason why glucosidases are usually considered as soil quality parameters [61].Oligopeptides, peptides, and/or free amino acids being hydrolysed by microbial activities in the soil thus supporting their growth and, consequently, their beneficial effects on plants providing nutrients.

In both cases, they would favour soil microbiota and its interaction with plants. It has been previously reported that protein hydrolysate stimulated growth of PGPR leading to increases in leaf chlorophyll and plant biomass [10] in lettuce. In fact, it is commonly accepted that soil microorganisms such as *Bacillus*, *Pseudomonas*, *Actinobacterial*, and *Lactobacillus* can improve plant performance by altering physiological and development processes resulting in a greater nutrient and water uptake as well as enhanced resilience against environmental stressors [1,62]. In addition, protein hydrolysis may result in the release of bioactive peptides. To date, only a few bioactive peptides have been identified which are responsible for the BS activity; thus, further investigation is necessary to clarify their mechanisms of action. Concerning the beneficial effects on in vitro plant cell cultures, as these are cultivated in sterile conditions, probably both Xylanase and β-Glucosidase provide more carbon and nitrogen sources directly to the cells through their degradation products, since the depolymerisation of enzyme-based biostimulants could not be excluded. Carbon and nitrogen are essential components of plant cells; it has been thoroughly demonstrated that amino acids derived from protein hydrolysates can directly modulate N assimilation by interfering with the expression of genes involved in primary C and N metabolism [4,5,63]. Further, exogenous amino acid application was found to influence plant hormone action [64].

## 5. Conclusions

This work demonstrated that some specific enzymes may act as BSs. In particular, among all the tested treatments, Xylanase and β-Glucosidase were the most promising, showing an increasing trend in fresh and dry weights and SPAD index of the lettuce plants, positively impacting on the accumulation of health-promoting compounds (i.e., total polyphenols, carotenoids, and ascorbic acid). These compounds are crucial at the plant cell level, taking part in adaptation to environmental changes but, most importantly, contributing to human health due to their antioxidative role. Specifically, among vitamins, ascorbic acid is one of the most relevant indicators of the nutritional quality of fruits and vegetables. Interestingly, as for the anti-oxidative power, Xylanase and β-Glucosidase, separately applied, were more effective than their combination, thus lacking an additive or a synergistic effect. However, when simultaneously applied they enhanced protein content in lettuce tissues possibly due to a metabolic effect. Indeed, even if a synergistic effect of Xylanase and β-Glucosidase as for the antioxidative properties could not be found, they might have induced protein synthesis providing carbon sources through their enzymatic activities. Furthermore, our data highlight the potentiality of the two biomolecules, separately tested, in improving plant cell proliferation and suggest that similar effects might be expected in in vivo systems. Indeed, based on the results achieved, the application of Xylanase and β-Glucosidase is a valuable strategy to increase carotenoid, polyphenol, and ascorbic acid contents, and consequently fresh yield. This aspect is of great interest in facing sustainable agricultural production under climate changes; therefore, further experiments will be performed to clarify the mechanisms of action of both molecules, to investigate their BS activity on different crops, and to evaluate their putative role in plant resilience and applicability in agricultural practices, including cellular agriculture.

## Figures and Tables

**Figure 1 biomolecules-13-01765-f001:**
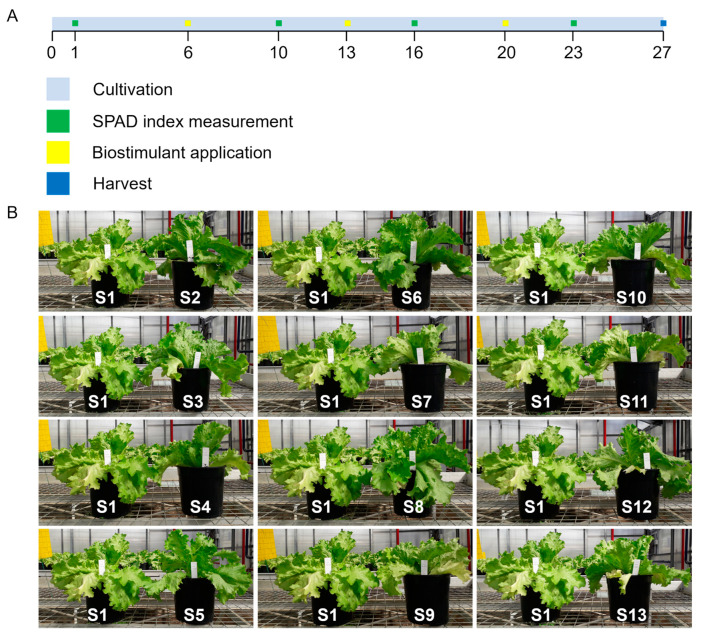
Timeline of lettuce plant cultivation (light blue bar). Green squares show the SPAD value detections; yellow squares show the three time points of the BS application; blue square shows the sampling time (**A**). Overview of lettuce plants at harvest time treated with different plant BSs in comparison with S1 controls (**B**). See Table 1 for detailed composition of different biostimulant solutions applied.

**Figure 2 biomolecules-13-01765-f002:**
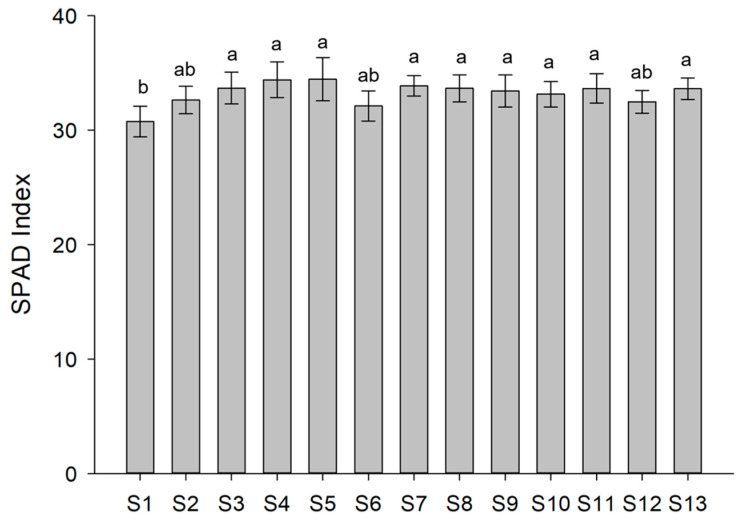
Chlorophyll content (SPAD index) of lettuce plants subjected to the different treatments. Values are expressed as mean of biological replicates (*n* = 15) ± SD. Different letters indicate significant differences using the Tukey test (*p* < 0.05; *n* = 15).

**Figure 3 biomolecules-13-01765-f003:**
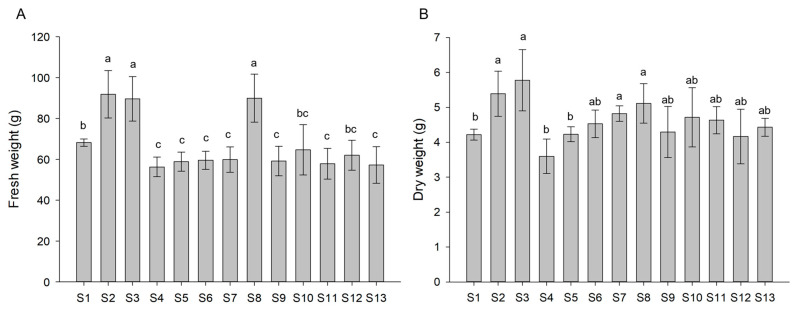
Fresh (**A**) and dry (**B**) weights of lettuce plants subjected to the different treatments. Values are expressed as mean of biological replicates (*n* = 10) ± SD. Different letters indicate significant differences using the Tukey test (*p* < 0.05; *n* = 10).

**Figure 4 biomolecules-13-01765-f004:**
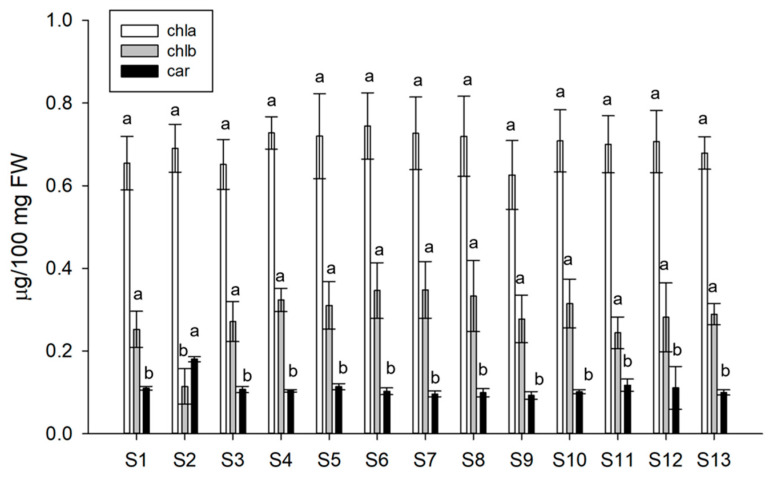
Pigment content of lettuce plants subjected to the different treatments. Values are expressed as mean of biological replicates (*n* = 5) ± SD. Different letters indicate significant differences using the Tukey test (*p* < 0.05; *n* = 5).

**Figure 5 biomolecules-13-01765-f005:**
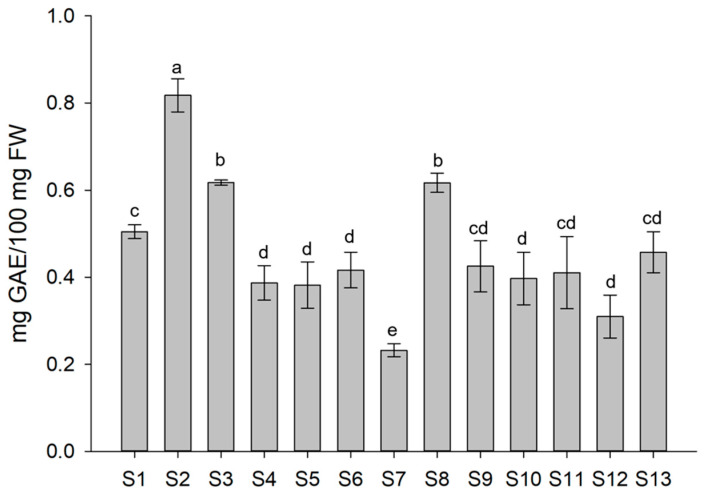
Total phenolic content (TPC) of lettuce plants subjected to the different treatments, expressed as gallic acid equivalents (GAEs) and measured using the Folin–Ciocalteu colorimetric assay. Each value represents the mean ± SD of five biological replicates. Different letters are significantly different between treatments at *p* < 0.05 by analysis of variance (ANOVA).

**Figure 6 biomolecules-13-01765-f006:**
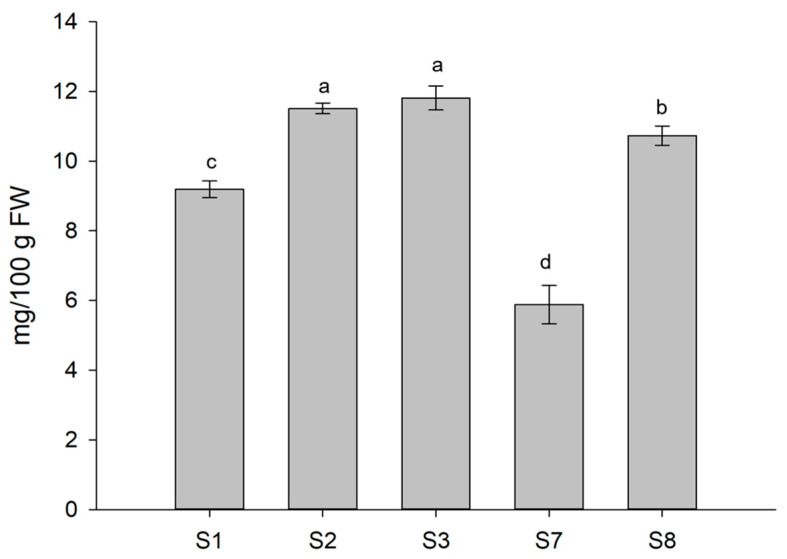
Ascorbic acid content (reduced form) of lettuce plants subjected to some of the different treatments. Different letters are significantly different between treatments at *p* < 0.05, using analysis of variance (ANOVA).

**Figure 7 biomolecules-13-01765-f007:**
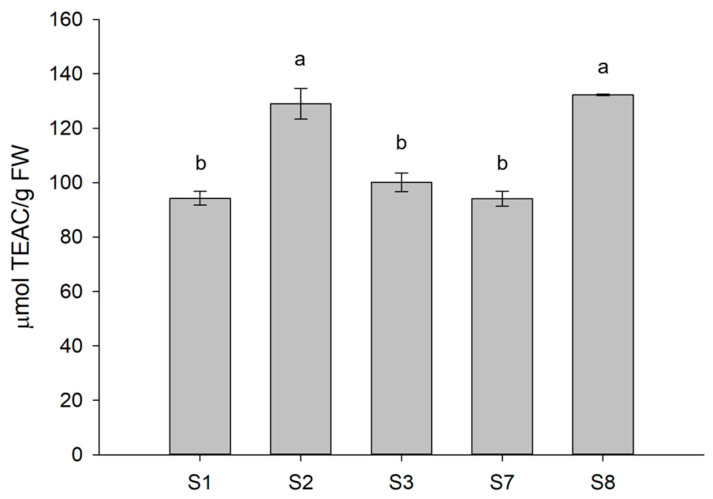
Antioxidant activity of lettuce plants selected among those subjected to the different treatments. Different letters are significantly different between treatments at *p* < 0.05, using analysis of variance (ANOVA).

**Figure 8 biomolecules-13-01765-f008:**
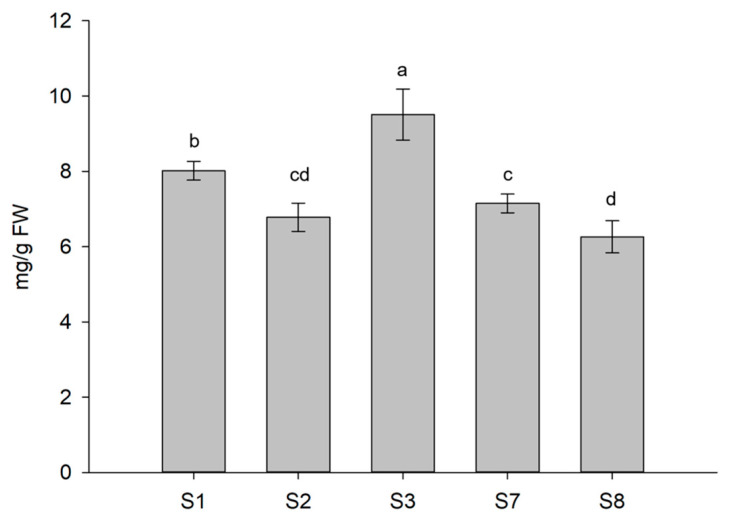
Total soluble proteins of lettuce plants selected among those subjected to the different treatments. Different letters are significantly different between treatments at *p* < 0.05, using analysis of variance (ANOVA).

**Figure 9 biomolecules-13-01765-f009:**
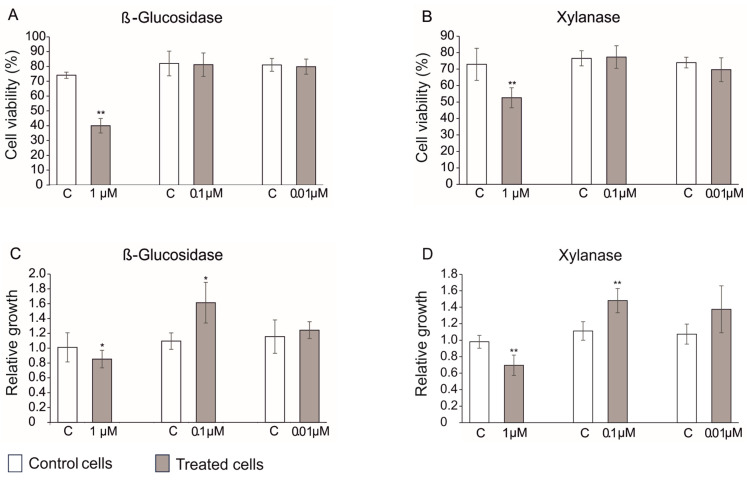
Impact of β-Glucosidase and Xylanase treatment on cell viability (**A**,**B**) and biomass growth of suspended kiwifruit cells (**C**,**D**). Viability was assessed using fluorescein diacetate and biomass growth was expressed as cell dry weight. Data expressed as means ± S.E. of three replicates. Asterisks indicate significance of differences at *p* < 0.05 (*) and *p* < 0.01 (**).

**Table 1 biomolecules-13-01765-t001:** Composition of solutions used as treatments.

Biostimulant Solution	PBS 1X	Xylanase	β-Glucosidase	Chitinase	Tramesan
S1	+	−	−	−	−
S2	−	+	−	−	−
S3	−	+	+	−	−
S4	−	+	−	−	+
S5	−	+	−	+	−
S6	−	+	+	−	+
S7	−	+	+	+	−
S8	−	−	+	−	−
S9	−	−	+	−	+
S10	−	−	+	+	−
S11	−	−	+	+	+
S12	−	−	−	+	−
S13	−	−	−	+	+

## Data Availability

Data are contained within the article.

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
