# Peer review of "Enzyme-Based Biostimulants Influence Physiological and Biochemical Responses of Lactuca sativa L."

_biomolecules, 2023, doi:10.3390/biom13121765_

Round 1

Reviewer 1 Report

Comments and Suggestions for Authors

This is an excellent series of experiments that not only may be of interest to other researchers in the field, bt also with a great potential to its immediate use in the culture of plants, and not oly lettuce.

Mi reccomendation is that it should be accepted as presented, unless some expert in english language may suggest corrections<

Author Response

We thank the reviewer 1 for the positive comments, we sincerely appreciated them.

Reviewer 2 Report

Comments and Suggestions for Authors

General comments:

The manuscript focuses on the Biostimulants (BSs) as natural materials (i.e., organic, inorganic compounds, and/or microorganisms) having beneficial effects on plant growth and productivity and being able to improve resilience/tolerance to biotic and abiotic stresses. The objective of this study was to investigate the effects of several biomolecules (i.e., Xylanase, β-Glucosidase, Chitinase, Tramesan), alone or in combinations, on lettuce plant growth and quality. The manuscript has scientific potential. A few points need to be addressed to improve the overall quality of the manuscript before publication.

The manuscript can be accepted after minor revisions. Moderate English editing and spell check is required.  A few typographical errors also need to be addressed.

Comments for authors:

Abstract: It should be presented with results rather than general statements.

Introduction: It is very brief. It should be elaborated to reflect the aim of this study.

L83: how many days-old seedlings were procured from the nursery?

L102: How the chlorophyll content was measured at the seedling stage?  At what stage, it was measured?

Figure 1: What represents S1, S2….? Mention in the caption.

Figure 2: What was the control? What a and b signifies. Also, the figures are of poor resolution.

Discussion: Can be improved with the latest references supporting the findings of the present study.

Conclusions: There should be no references in the conclusion section. It should be well written and highlight the present study's significant findings.

Comments on the Quality of English Language

Moderate English editing and spell check is required.  A few typographical errors also need to be addressed.

Author Response

Comments for authors:

Abstract: It should be presented with results rather than general statements.

R: We agree with reviewer for this suggestion, thus we deleted the first two sentences related to general statement.

Introduction: It is very brief. It should be elaborated to reflect the aim of this study.

R: Accordingly to reviewer’s suggestion, we expanded the introduction, giving a particular attention to the aim of this study (Lines 39-42, 44-45, 55-57, 69-72, 79-85)

L83: how many days-old seedlings were procured from the nursery?

R:The plants received from the nursery were about 3 weeks old, we have added this info in the 2.1 section of materials and methods (Line 98)

L102: How the chlorophyll content was measured at the seedling stage?  At what stage, it was measured?

R:In section 2.3 of Materials and methods we described the two different methods used for Chlorophyll measurements. During the time frame of the experiments, we used SPAD meter for chlorophyll determination since it is a non-destructive method. The measurements were performed weekly, three days after BS treatments. At the end of the experiments, seven days after the last treatment (27 DAT), lettuce plants were harvested and used for both biomass determination and biochemical analyses. For pigments quantification (chlorophyll a and b, and carotenoids), we used spectrophotometric measurements accordingly to the methods described by Lichtenthaler (1987), as reported in the section 2.5. Therefore, to better clarify our experimental procedures, we have added more details about plant stage and experiments duration in section 2.3 of Material and Methods (Lines 121-122).

Figure 1: What represents S1, S2....? Mention in the caption.

R: Accordingly to reviewer’s suggestion and to avoid a long and complex description of all treatments, we have added in the caption of Figure 1 the sentence “See Table 1 for detailed composition of different biostimulant solutions applied” (Lines 224-225).

Figure 2: What was the control? What a and b signifies. Also, the figures are of poor resolution.

R: We have added in the R1 version of the manuscript high resolution images for all figures.

S1-treated plants correspond to control group. S1 solution contains only PBS buffer 1X. As described in the figure legend, the letters on the bars indicate statistical analysis performed by One way Anova method, thus different letters means statistical significance among treatments.

Discussion: Can be improved with the latest references supporting the findings of the present study.

R: Accordingly with reviewer’s suggestions, we improved Discussion section by adding new references [20, 35, 36, 47, 48, 55, 62] that support our findings.

Conclusions: There should be no references in the conclusion section. It should be well written and highlight the present study's significant findings.

R: We have modified the conclusions according to reviewer’s suggestion.

Reviewer 3 Report

Comments and Suggestions for Authors

The only issue I have with work is in the  2.1. Plant material and growth conditions must be improved significantly. 

How often do they water the treatments? 

Do they use water, a nutritive solution or a fertilizer?

Do they have the same light intensity or temperature (Max-Min)?

The rest of the paper is more than fine.

Author Response

The only issue I have with work is in the 2.1. Plant material and growth conditions must be improved significantly. 

R:We thank the reviewer for suggestions, we modified the section 2.1 of Materials and Methods as requested (Lines 98, 101-104)

How often do they water the treatments?

R: We have added the daily irrigation in the text (Lines 102-104)

Do they use water, a nutritive solution or a fertilizer?

R:We have specified in the text that we irrigated with tap water daily and treated with 50 ml of BSs weekly (lines 102-104)

Do they have the same light intensity or temperature (Max-Min)?

R: We did not measure the light intensity, we have added the max and min temperatures (lines 101)

The rest of the paper is more than fine.